# Vestibular Anatomic Localization of Pain Sensitivity in Women with Insertional Dyspareunia: A Different Approach to Address the Variability of Painful Intercourse

**DOI:** 10.3390/jcm9072023

**Published:** 2020-06-27

**Authors:** Ahinoam Lev-Sagie, Osnat Wertman, Yoav Lavee, Michal Granot

**Affiliations:** 1Faculty of Medicine, Hebrew University of Jerusalem, Israel, and Clalit Health Organization, 12 Faran St, Jerusalem 9780214, Israel; 2School of Social Work, Faculty of Social Welfare and Health Sciences, University of Haifa, Haifa 3498838, Israel; wertmanosnat@gmail.com (O.W.); ylavee@univ.haifa.ac.il (Y.L.); 3Faculty of Social Welfare and Health Sciences, University of Haifa, Haifa 3498838, Israel; granot@research.haifa.ac.il

**Keywords:** provoked vestibulodynia, insertional dyspareunia, vestibular tenderness, Q-tip test

## Abstract

The pathophysiology underlying painful intercourse is challenging due to variability in manifestations of vulvar pain hypersensitivity. This study aimed to address whether the anatomic location of vestibular-provoked pain is associated with specific, possible causes for insertional dyspareunia. Women (*n* = 113) were assessed for “anterior” and “posterior” provoked vestibular pain based on vestibular tenderness location evoked by a Q-tip test. Pain evoked during vaginal intercourse, pain evoked by deep muscle palpation, and the severity of pelvic floor muscles hypertonicity were assessed. The role of potential confounders (vestibular atrophy, umbilical pain hypersensitivity, hyper-tonus of pelvic floor muscles and presence of a constricting hymenal-ring) was analyzed to define whether distinctive subgroups exist. Q-tip stimulation provoked posterior vestibular tenderness in all participants (6.20 ± 1.9). However, 41 patients also demonstrated anterior vestibular pain hypersensitivity (5.24 ± 1.5). This group (circumferential vestibular tenderness), presented with either vestibular atrophy associated with hormonal contraception use (*n* = 21), or augmented tactile umbilical-hypersensitivity (*n* = 20). The posterior-only vestibular tenderness group included either women with a constricting hymenal-ring (*n* = 37) or with pelvic floor hypertonicity (*n* = 35). Interestingly, pain evoked during intercourse did not differ between groups. Linear regression analyses revealed augmented coital pain experience, umbilical-hypersensitivity and vestibular atrophy predicted enhanced pain hypersensitivity evoked at the anterior, but not at the posterior vestibule (R = 0.497, *p* < 0.001). Distinguishing tactile hypersensitivity in anterior and posterior vestibule and recognition of additional nociceptive markers can lead to clinical subgrouping.

## 1. Introduction

Among sexual dysfunctional conditions, insertional dyspareunia is a general term describing extreme discomfort or pain felt at the entrance of the vagina during vaginal penetration or penetration attempt. The severity can range from a total inability to tolerate penetration, to a feasible but painful experience. The provoking stimulus is typically the attempted entry and movement of the penis, but it can also be a finger, dildo, speculum or tampon. As this disorder may involve different etiologies and characteristics, both physical and psychological, it has been investigated for many years by various disciplines (pain, gynecology, sexology, psychiatry etc.), and was therefore, described using multiple names and varied definitions. For example, it is referred to as “genito-pelvic pain/penetration disorder” by the DSM-5 [1], “introital dyspareunia” and “superficial dyspareunia”. Insertional dyspareunia may result from various causes, including infections and inflammation. As insertional dyspareunia may result from involvement of the vulvar vestibule, a part of the vulva located adjacent to the vagina (Figure 1), as well as from severe contraction of the pelvic floor muscles encircling the vaginal entrance, this general term also encompasses the diagnoses of provoked vestibulodynia (PVD) and vaginismus, respectively.

PVD is considered the most common form of premenopausal dyspareunia, affecting 12–21% of women [2,3,4]. PVD refers to pain confined to the vulvar vestibule, in response to contact or pressure, lasting at least three months [5], and is a subset of vulvodynia (vulvar pain without clear identifiable cause). PVD is attributed to various etiologies including abnormalities in the vestibular mucosa [6,7], pelvic floor hypertonicity [8,9], alteration of peripheral and/or central pain modulation processing [10], as well as different mediating factors, including inflammatory, hormonal, genetic, psychosocial, etc., [10,11,12,13]. As a result, it was suggested that PVD is not one disease but a constellation of symptoms of several (sometimes overlapping) disease processes [5] with significant variation in the location, intensity, duration, and sensory quality of pain. Although extensive research expanded the understanding of PVD pathophysiology, the diagnostic criteria of PVD did not advance accordingly, and still relies on those suggested more than 30 years ago [14], including severe provoked pain on vestibular touch, marked tenderness to cotton swab palpation of the vulvar vestibule and exclusion of other, identifiable causes [5]. PVD is considered a diagnosis of exclusion by definition, however, which “identifiable” causes must be ruled out before this diagnosis can be made are not clearly defined. Therefore, it is common that diagnosing PVD varies significantly between providers, based on the background, knowledge, examination skills and experience of the examiner, rather than on formal criteria.

In spite of extensive research of the possible etiologies for PVD, clinical practice has not advanced respectively, as the relative contribution of these factors to pain hypersensitivity remains poorly understood. As personalized treatment according to etiology and associated PVD factors is not yet available, a “trial and error” therapeutic approach [15] is common practice. An alternative approach was suggested by some authors [11,16,17], based on the observation that location and characteristics of vestibular pain intensity and quality vary among patients [17,18,19]. As these clinical observations reveal that some patients diagnosed with PVD present with an enhanced pain response to tactile stimulation in the entire vestibule, while others demonstrate pain hypersensitivity only at the posterior (lower) vestibule, these authors suggested that the presence or absence of anterior vestibular mucosal sensitivity represents different PVD subgroups and may aid in differentiation of PVD-etiology [11,16,17]. However, data to support this theory are scarce. Assuming that grouping together all women with vestibular tenderness overlooks potentially crucial differences and may result in inadequate therapeutic intervention, we aimed to explore whether distinctive presentation and location of vestibular tenderness may further identify different causes for insertional dyspareunia.

Given that subgrouping women with vestibular tenderness may allow for personalized treatment, and consequently, improved outcomes, the present study was conducted to address whether the specific location of vestibular provoked pain reflects various insertional dyspareunia subgroups.

Since “PVD” is defined in the literature as vestibular pain of unknown etiology and the aim of this study was to recognize possible neglected causes, we used the terms “insertional dyspareunia” and “vestibular-tenderness” instead. The present study was based on the role of possible confounders that explain vestibular pain hypersensitivity in insertional dyspareunia [20,21,22,23,24], as well as on the previously proposed algorithm [16], which distinguishes between subgroups by incorporating localization of vulvar tenderness on examination along with the patient’s history [11,17]. We assumed that a framework of classification into subgroups is required to address the distinct etiologically based typology of insertional dyspareunia.

Specific goals were to define vestibular tenderness subgroups by: (1) vestibular pain location (anterior vs. posterior) as induced by experimental pain (Q-tip test), and (2) identification of additional pain-related features that may contribute to the development and expression of vestibular pain hypersensitivity in insertional dyspareunia.

## 2. Materials and Methods

### 2.1. Study Population

This study is part of a prospective, longitudinal, cohort-based research project that evaluated clinical characteristics, pain-related psychosexual and cognitive features, various pain measures, as well as response to treatment in patients diagnosed with vestibular tenderness and insertional dyspareunia. The cohort consisted of women evaluated between November 2016 and January 2019 in a specialized clinic for vulvovaginal disorders. Patients who fulfilled the diagnostic criteria of PVD [5], and were willing to participate in the study, were enrolled and evaluated in a standard manner (see below). The study was approved by the local Institutional Review Board of Clalit Health Organization (COM1-16-89) and Haifa University (Identification code: 353/16), and was conducted in accordance with the Declaration of Helsinki. I Informed consent was obtained from all patients after receiving a detailed explanation, before participation in the study. For the purpose of the current study, data obtained in the first examination and assessment session were analyzed. Although we acquired multiple parameters, including different experimental variables (tampon test, algesiometer), psychosocial data and various clinical parameters, only clinical norms for diagnosis were included in this manuscript, which can be reproduced by an examining caregiver in the clinic, in order to provide practical diagnostic parameters.

Inclusion criteria included: (1) a history of ≥3 months of vulvar pain suggestive of PVD (pain during vaginal penetration and/or pain with tampon insertion) [5]; (2) self-reported sexual pain intensity ≥3 in the 0–10 Visual Analogue Scale (VAS) or complete avoidance of penetrative intercourse due to pain severity; (3) tenderness localized within the vestibule on examination, with pain scores > 1 in response to Q-tip stimulation; (4) age 18–45.

Exclusion criteria included: (1) an identifiable cause for the pain, such as vulvovaginitis, dermatitis, skin disorder etc.; (2) major health conditions (cardiovascular, diabetes etc.,); (3) diagnosis of an unstable psychiatric disorder; (4) pregnancy or lactation; (5) medication effecting neuromodulation and (6) diagnosis of spontaneous (i.e., pain arising without any provoking physical contact) or mixed vulvodynia (i.e., a combination of spontaneous vulvodynia and pain provoked by physical contact).

### 2.2. Patient Evaluation Measures

All patients were evaluated in a standard manner by the same gynecologist, a specialist in vulvovaginal disorders (the first author).

Medical and gynecological examination: after obtaining a detailed medical, obstetric, and gynecological history, a vulvovaginal examination was performed. This included vaginal pH measurement, saline and 10% potassium hydroxide microscopy, yeast cultures and sexually transmitted infections (STIs) screening. In addition, localized vestibular atrophy, characterized by vestibular (but not vaginal) mucosal thinning, dryness and erythema was assessed. This particular vestibular atrophy is not referred to as “vaginal atrophy”, is often ignored and thus, its description is usually absent from clinical assessment. Patients were asked to provide dyspareunia history (primary/secondary), duration of dyspareunia, and current or prior use of systemic hormonal contraception (HC): oral contraceptives, transdermal patch, or vaginal ring.Pain evoked during vaginal intercourse: Self-report of pain intensity ratings experienced during sexual intercourse were assessed using a 0–10 Visual Analogue Scale (VAS), with 0 representing no pain and 10 being the worst possible pain.Assessment of vestibular pain [25]: Vestibular tenderness was assessed by the Q-tip test, using a moistened cotton-tip applicator and touching the vestibule in four defined points (2, 4, 8 and 10 o’clock—Figure 1), with an interval of 5 s between each stimulus. The Q-tip test was performed twice, first to localize vestibular tenderness at each point (yes/no) and secondly, to quantify pain intensity using a Numeric Pain Scale (NPS) ranging from 0 to 10 at each point, with 0 corresponding to no pain and 10 being the worst possible pain.

Pain rating in response to deep muscle palpation: patients were requested to report pain intensity using a 0–10 NPS, in response to pressure applied bilaterally to the puborectalis muscles with the examiner’s index finger.Pelvic floor muscle hypertonicity: the physician’s impression of hypertonicity (mild, moderate and severe) of the pelvic floor musculature was measured by applying pressure with the examiner’s index finger bilaterally to the puborectalis muscles.Assessment of rigid/constricting hymenal ring: This was done by placing 2 fingers at the introitus and stretching the hymenal ring (Figure 1) laterally [26,27], avoiding pressing or stretching of the underlying muscles. If insertion of two fingers was impossible due to obliterating hymenal tissue (but not contraction of the muscles or vaginismus), or if this hymenal-ring stretching provoked pain similar to the pain experienced by the patient with penetration and the physician identified a thick/rigid hymen, the patient was reported to have a “constricting hymen”.Umbilical hypersensitivity: Umbilical tenderness was assessed by a dry cotton-tip applicator by touching it gently and asking the patient to report hypersensitivity (yes or no). Given the common endodermal embryological origin of the vestibular mucosa and the umbilicus, hypersensitivity to touch in this location was considered to be a possible representative of congenital vestibular neuroproliferation [28].Level of desire and vaginal lubrication were assessed by calculation of the relevant domains in the Female Sexual Function Index, which was completed by the participants.

### 2.3. Allocation into the Anterior and Posterior Vestibular Tenderness Groups

Pain scores > 1 in response to Q-tip stimulation at the 2 and 10 o’clock vestibular locations (Figure 1) were defined as “anterior vestibule” hypersensitivity and were calculated as the average of measurements obtained at these two points. “Posterior vestibule” hypersensitivity was defined as the average of the pain scores at 4 and 8 o’clock locations (Figure 1). Women were divided into 2 subgroups based on the location of the hypersensitivity. Patients with both anterior and posterior vestibular tenderness were defined as having “circumferential vestibular tenderness”, while those with posterior-only pain sensitivity were defined as having “posterior-only vestibular tenderness”.

### 2.4. Statistical Analyses

Statistical analyses were performed with SPSS version 23 (SPSS Inc., Chicago, IL, USA). Normal distribution was tested using the Shapiro–Wilk procedure. ANOVA was used to determine whether the subgroups of vestibular tenderness were different in relation to experimental or clinical pain measures. Variables with a non-normal distribution were analyzed using non-parametric tests; namely the Wilcoxon-Signed Ranks test for the comparison of two dependent groups and the Mann–Whitney U-test for the comparison of two independent groups. Correlation of non-normally distributed variables was carried out using Spearman rank tests and the association between binary variables was analyzed using Chi-square tests. Linear regression analysis was used to identify predictors for enhanced vulvar pain provoked at the anterior or posterior vestibule. Statistical significance was set at *p* < 0.05.

## 3. Results

### 3.1. Patients’ Characteristics

Of the 154 patients who were evaluated for insertional dyspareunia, 26 were excluded for the following reasons: 10 had mixed vulvodynia, five were breastfeeding, seven were receiving medication effecting neuromodulation and four had concurrent vulvar dermatosis. Of the 128 women who met inclusion criteria, 113 consented to participate. Patients’ sociodemographic characteristics are presented in Table 1.

### 3.2. Characteristics of Circumferential Vs. Posterior-Only Vestibular Tenderness Hypersensitivity

All women reported provoked pain sensation in response to Q-tip stimulation at the posterior vestibule; 41/113 patients (36.3%) also reported vestibular-hypersensitivity as expressed by provoked pain >1 in response to the Q-tip test at the anterior vestibule as well and, thus, were defined as the **Circumferential vestibular tenderness group**. The remaining 72 patients, who experienced no pain or reported NPS ≤ 1 in the Q-tip test at the anterior vestibule, were defined as the **Posterior-only vestibular tenderness group**. Interestingly, group comparison for Q-tip pain ratings obtained from the posterior vestibular points showed significantly higher pain scores in the Circumferential vestibular tenderness group as compared to the Posterior-only vestibular tenderness group (mean 6.84 ± 1.9 vs. 5.85 ± 1.8, median 7.0 vs. 5.9, Mann–Whitney, Z = −2.77, *p* = 0.007). However, no differences between the two groups were observed regarding pain intensity evoked by deep muscle palpation (5.62 ± 2.3 vs. 6.08 ± 1.9, *p* = 0.233) or perceived pain intensity during vaginal intercourse (8.14 ± 1.4 vs. 7.71 ± 1.8, *p* = 0.198). In addition, age and duration of dyspareunia symptoms were not different between groups.

### 3.3. Characteristics of Vestibular-Hypersensitivity in the Circumferential Vestibular Tenderness and the Posterior-Only Vestibular Tenderness Groups

To further explore the meaning of pain localization in the context of possible pathophysiology, the role of additional confounders, possibly associated with vestibular augmented pain hypersensitivity, were examined. Accordingly, group comparisons for the appearance of possible alterations that may contribute to the enhanced pain response were performed. The results are presented in Table 2 and show no differences between the two groups (Circumferential or Posterior-only) regarding HC use. In the entire sample, a greater presence of vestibular mucosal atrophy was associated with current use of HC. More specifically, 27/42 (64.3%) women who demonstrated vestibular atrophy were using HC, whereas 17/67 (25.4%) of those with no such atrophy reported using HC (Chi = 16.2, *p* < 0.001, four were missing responses regarding the use of HC). Furthermore, presence of vestibular atrophy was associated with pain location, as 26/41 (63.4%) of the women in the Circumferential vestibular tenderness group demonstrated vestibular atrophy, whereas only 15/72 (20.8%) in the Posterior-only group had vestibular atrophy (Chi = 17.6, *p* < 0.001), thus, leading to the hypothesis that the Circumferential vestibular tenderness group comprised patients with more than one possible etiology for vestibular pain. Therefore, to identify possible confounders that could contribute to enhanced pain in the anterior vestibular location, the role of umbilical pain hypersensitivity in response to touch was examined. This was chosen because it was suggested to be an indicator of augmented pain response representing neuroproliferation [28]. In the Posterior vestibular tenderness group, 13/72 (18.1%) of the women had umbilical hypersensitivity, while 19/41 (46.3%) women in the Circumferential vestibular tenderness group had umbilical hypersensitivity (Chi = 10.3, *p* = 0.001).

The Posterior-only group was characterized by higher scores of pelvic floor hypertonicity as graded by the examining physician. Thirty-nine out of 58 (67.2%) women with severe hyper-tonus of the pelvic floor muscles were in the Posterior-only group as compared to 19/58 (32.8%) in the Circumferential vestibular tenderness group (Chi = 8.77, *p* = 0.012). The difference in the degree of the pelvic floor hypertonicity (mild, moderate or severe) is shown in Figure 2.

Next, we evaluated the possible role of pelvic floor hypertonicity, as well as of rigid hymen in the context of the vestibular tenderness-groups and vestibular pain location. Interestingly, in the Posterior-only vestibular tenderness group, a portion of the patients demonstrated rigid hymen (Table 2). Thus, similar to the subgrouping performed in the exploration of the Circumferential vestibular tenderness group, the Posterior-only group were divided into two sub-populations—one was defined by the presence of a hymenal-constriction ring as an observed anatomical/functional finding, and the other subgroup comprised women who did not demonstrate this finding.

### 3.4. Construction of the Four Vestibular Tenderness Subgroups

Based on localization of pain, as well as on the abovementioned features that characterized each subgroup, and on the previously proposed algorithm for PVD classification [16,19,25], the sample was subdivided into four different subgroups (Figure 3); patients with Circumferential vestibular tenderness (*n* = 41) were divided into two subgroups. Women with vestibular atrophy and current or past usage of HC at the time of symptoms appearance, possibly experiencing hormonally mediated vestibular sensitivity, were classified as **Hormonal-mediated** vestibular tenderness (n = 21). The remaining 20 patients who had Circumferential vestibular tenderness, lacking vestibular atrophy, were termed **Augmented anterior** vestibular tenderness. The 72 women who had Posterior-only vestibular pain were subdivided into two subgroups based on the existence of a hymenal constriction-ring, representing rigidity of the connective tissue. Accordingly, 37 women with hymenal-rigidity were termed **Hymenal**-vestibular tenderness, and the remaining 35 patients were termed **Hypertonic** vestibular tenderness.

#### 3.4.1. The Distinctive Characteristics of the Four Subgroups

Whereas no difference was observed between the Circumferential vestibular tenderness group and the Posterior-only group in the incidence of primary (painful coitus from the first vaginal penetration) or secondary dyspareunia appearance, the Chi-square test showed that dividing the patients into these four subgroups revealed a significant association between belonging to any of the four groups and the diagnosis of primary or secondary vulvar dyspareunia (Chi = 28.1, *p* < 0.001). It was found that 10/21 (47.6%) of the Hormonal-vestibular tenderness subgroup reported primary dyspareunia, whereas 17/20 (85%) of the Augmented anterior-vestibular tenderness patients were diagnosed with primary dyspareunia (Chi = 33.0, *p* < 0.001). A similar distinction was observed in the Hymenal-vestibular tenderness subgroup, in that 35/37 (94.6%) of the patients had primary dyspareunia, whereas in the Hypertonic-vestibular tenderness subgroup, 13/35 (37.1%) of the women were diagnosed with primary dyspareunia (Chi = 21.8, *p* < 0.001). The two subgroups of women with Circumferential vestibular tenderness were also different in relation to patient complaint of vaginal dryness (i.e., lack of lubrication). A higher incidence of vaginal dryness was reported in the Hormonal-vestibular tenderness subgroup, 16/20 women (80%, one missing report) as compared to 10/20 (50%) of women in the Augmented anterior-vestibular tenderness subgroup (Chi = 5.96, *p* = 0.015). Additionally, significantly more patients in the Hormonal-vestibular tenderness subgroup reported decreased desire, 19//20 (95%, one missing), whereas the incidence of patients with decreased desire in the Augmented anterior-vestibular tenderness group was significantly lower, 11/20 (55%) (Chi = 8.53, *p* = 0.003). However, no significant difference in the incidence of decreased desire was found between the two subgroups of the Posterior-only vestibular tenderness group.

#### 3.4.2. Four Group Comparisons of Experimental Provoked Pain Measures

The results of the ANOVA for pain intensity scores, as assessed by the Q-tip tests in each subgroup at the anterior and posterior vestibule, are presented in Figure 4.

Women in the Hormonal-vestibular tenderness and the Augmented anterior-vestibular tenderness subgroups, comprising the Circumferential vestibular tenderness group, reported significantly higher provoked pain ratings at the anterior vestibule as compared to the Posterior-only two subgroups (Hymenal-vestibular tenderness and Hypertonic-vestibular tenderness) (F = (_3112_) = 66.8, *p* < 0.001). Post hoc (Tukey HSD) analysis showed no significant differences within groups (the Hormonal-vestibular tenderness vs. Augmented anterior-vestibular tenderness and the Hypertonic-vestibular tenderness vs. Hymenal-vestibular tenderness). As for the pain ratings evoked by the Q-tip stimulation delivered to the posterior vestibule, a trend for similar characteristics was obtained, but did not reach a significant level (F = (_3112_) = 2.60, *p* < 0.056). Notably, no difference was observed between the four groups in relation to the reported pain experience during vaginal intercourse or intensity of pain evoked by deep muscle palpation.

### 3.5. Prediction of Augmented Pain Hypersensitivity at the Anterior Vestibule

In order to define factors associated with Q-tip pain intensity as experienced at the anterior-vestibule location, a linear regression analysis was conducted. This model (R = 0.497, *p* < 0.001) showed that for the entire sample, higher pain experienced during intercourse, the existence of umbilical hypersensitivity, as well as vestibular atrophy, but not the degree of muscle hypertonicity or deep palpation pain, are associated with enhanced pain sensitivity at the anterior vestibule (Table 3).

### 3.6. Prediction of Augmented Pain Hypersensitivity at the Posterior Vestibule

To further confirm our hypothesis about the key impact of pain localization in the understanding of vestibular tenderness pathophysiology, the same regression model was performed to explain pain variability obtained in response to Q-tip stimulation of the posterior vestibular area (R = 0.562, *p* < 0.001). This analysis revealed that only self-reported pain intensity during intercourse was associated with enhanced posterior vestibular pain ratings (Table 4).

## 4. Discussion

Given that a significant portion of patients presenting with vestibular tenderness and diagnosed with PVD do not benefit from the treatments recommended by professional guidelines [15,29], this study was conducted to better address the possible role of less explored factors that may be involved in the development and manifestation of vestibular tenderness.

The current terminology of vulvar pain, revised in 2015, defines “vulvar pain caused by a specific disorder”, applying to conditions for which a cause can be clearly identified (such as yeast infection, skin disease, vulvovaginal atrophy due to estrogen deficiency etc.,) and “vulvodynia”, vulvar pain without clear identifiable cause, which may have potential associated factors. The addition of “potential associated factors” is considered the main difference between the 2015 terminology and the 2003 terminology [5]. Some of the associated factors mentioned in the 2015 consensus are hormonal factors, pelvic muscle overactivity, neuroproliferation and structural defects, suggesting that they may be clinically prominent, and stressing that treatment should be selected according to possible associated factors. Additionally, it is traditionally recommended that vestibular tenderness, assessed using the Q-tip test, should evaluate pain hypersensitivity in several vestibular locations (i.e., 2, 4, 8 and 10) [15]. However, the practical significance of this localization with regard to possible associated factors has not been studied, nor has its clinical implication been defined.

The importance of vestibular pain localization, therefore, lies in the possibility of subgrouping patients in order to provide personalized treatment. Therefore, the combined effect of vestibular location of pain hypersensitivity, as well as other key features linked with pain alteration observed during the assessment process, which are often less assessed or overlooked (such as vestibular atrophy and hymenal constricting ring), were investigated. Our approach was based on combining well-accepted diagnostic parameters [14,25], previous observations of hypersensitivity localization within the vestibule [17,18,19] and other clinical findings known to contribute to pronociceptive response [10,11,12].

Although pain evoked by vaginal intercourse did not differ between groups, the finding that only part of the participants demonstrated vestibular circumferential pain hypersensitivity in response to controlled Q-tip stimulation, as compared to those with posterior-only vestibular hypersensitivity, emphasizes the notion that pain localization represents a differential manifestation of mucosal pain hypersensitivity in the vestibular regions. This concept has already been proposed [19,25,30,31,32], however, it is not confirmed by empirical research. This investigational approach was the basis for the search for further distinctive mechanisms within the Circumferential and the Posterior-only subgroups. Women with circumferential vestibular hypersensitivity were characterized by the presence of vestibular mucosal atrophy (localized to the vestibule without vaginal atrophic changes and associated with greater use of HC) or enhanced umbilical tactile pain hypersensitivity. However, women with posterior-only hypersensitivity were characterized by the existence of a rigid hymen or by severe pelvic floor hypertonicity. These differences support the notion of etiological confounders that affect pain localization as well as clinical symptoms.

Previous reports suggested that HC usage can cause morphological alterations of vestibular mucosa [22,33,34], which may result in dyspareunia [35,36,37,38,39]. In fact, most women who use HC do not report dyspareunia [40,41], however, hormonal status might cause an alteration of nociceptive reactivity in predisposed women, as observed in this cohort, where only one distinctive subgroup of women was possibly more vulnerable to the effect of HC [22,38,42,43]. Hormonal influence may present as vestibular atrophy and dyspareunia due to local changes in the vestibule and/or altered pain modulation at the systemic level [33,37,38,44]. The fact that those women demonstrated anterior vestibular pain highlights the major role of pain localization in identifying particular subgroups of patients with vestibular tenderness and insertional dyspareunia. This assertion is supported by the observation that most women in this subgroup experienced secondary dyspareunia. Thus, patients with hormonally mediated vestibular tenderness had some pain free periods before the use of HC, preceding the development of atrophy and/or enhanced pain sensitivity.

Vestibular pain in women with circumferential hypersensitivity, but without atrophy, may be attributed to vestibular neuroproliferation and/or pronociception. This assumption is supported by reports that demonstrated, based on immunohistochemical methods, hyperinnervation of nociceptors in the vestibular mucosa in PVD patients as compared to controls [7,20,21,45,46,47,48,49], perhaps resultant of congenital or acquired neuronal sensory alteration [11]. Embryologically, the vestibule is of endodermal origin, derived from the urogenital sinus. It was suggested that endodermal neuroproliferation may present concurrently in tissues derived from the same origin, such as the vestibule and the umbilicus. Indeed, umbilical hypersensitivity to touch was observed mainly in primary-PVD women as compared to those with secondary PVD and controls [28], but in this study, the role of pain location was not investigated. Accordingly, the Augmented anterior vestibular tenderness subgroup was characterized by circumferential provoked pain together with umbilical hypersensitivity, and primary onset of dyspareunia, observed in 90% of women in this subgroup. Unfortunately, biopsies were not performed in this study, and the neuroproliferation hypothesis should be further confirmed using advanced methods in future studies.

Patients in the Hymenal-vestibular tenderness subgroup demonstrated a rigid hymenal-ring, which might explain the pain noted during penetration. This is supported by the findings that most of these women reported primary dyspareunia, i.e., pain evoked during the first intercourse. Interestingly, a rigid hymenal ring, as a distinctive etiology for vestibular tenderness, has not yet been reported. It should be noted that hymenal remnants are frequently observed, generally painless in stretching during a gynecological exam. In addition, an inelastic hymen may cause constriction at the vaginal opening, which may result in difficult and painful penetration, due to direct stretching that provokes pain and possibly repeated tears [26,27]. In order to prevent unnecessary pain, it is often suggested that the manual examination be performed with one finger inserted through the hymen, without touching the vestibule, instead of the usual two fingers [25]. This method may cause overlooking hymenal-ring rigidity and may miss its possible contribution to insertional dyspareunia and coexistence with vestibular tenderness.

The remaining patients with posterior-only vestibular tenderness (termed Hypertonic- vestibular tenderness), demonstrated a tendency for enhanced pain ratings evoked by deep palpation of the pelvic floor muscles and more severe muscle hypertonicity. It was previously hypothesized that trigger points within the pelvic floor muscles may refer pain to the vestibular posterior region [8,50]. Alternatively, hypertonicity of the muscles that insert at the posterior vestibule (pubococcygeus, puborectalis, and superficial transverse perineum) might provoke pain in the posterior region due to altered neurodynamics, neural and tissue hypoxia [11]. Whether these women represent a distinctive posterior-only vestibular tenderness group, characterized by hypertonicity and augmented response to deep soma pressure applied to the pelvic floor muscles is still unknown. This manifestation may be secondary to enhanced pain sensitivity at the vestibule resultant from hypertonicity, or that the expected painful coitus among these patients leads to hyper-tonus as a ‘protective mechanism’ and consequently, to dyspareunia [8,9,51]. Our findings support the notion that women with ‘non-structural’ features of pain hypersensitivity differ from patients with posterior-only vestibular tenderness with an observed structural etiology, i.e., ‘narrow path’ of the vagina, that can explain the experience of painful coitus. The fact that dyspareunia among the Hypertonic-vestibular tenderness subgroup was mainly reported as secondary dyspareunia, supports the notion that a protective response to anticipated painful coitus characterizes these patients. In this regard, our findings shed more light on the importance of the patients’ history. More patients reported primary dyspareunia among the Augmented anterior-vestibular tenderness and Hymenal-vestibular tenderness groups, which may represent more ‘congenital conditions’, such as tactile pain hypersensitivity, possibly caused by congenital neuroproliferation in some patients and a rigid hymen in others, thus, explaining why painful coitus was experienced from the first intercourse experience. This may also explain why in the Hormonal-vestibular tenderness subgroup, a portion of the women who used HC before initiation of intercourse present as primary dyspareunia, while those who initiated HC following intercourse debut were characterized as secondary dyspareunia, implying that these women were pain free before the use of HC. These findings further support the notion that these are distinctive subgroups of vestibular tenderness, highlighting the need to perform a thorough history taking during the assessment.

However, subgroup placement did not affect the intensity of pain reports of dyspareunia during vaginal intercourse. These findings emphasize the value of attaining information based on multifaceted measures for provoked vestibular pain, e.g., experimental and ‘real-life’, which better reflect the context, settings and the effect of the proposed etiological mechanism as expressed by the location of vestibular pain.

Several limitations should be acknowledged, including, mainly, the absence of a control group and evidence of neuroproliferation (biopsies). Additionally, the small number of patients in the Augmented Anterior-vestibular tenderness and Hormonal-vestibular tenderness subgroups affect the strength of the findings. The lack of a standardized, objective method to assess muscle tonicity should be also acknowledged, although the data obtained for this measure have shown to have good intra-examiner consistency [31]. Furthermore, it should be noted that this study focused on possible etiological explanations for vulvar pain not related exclusively to PVD.

## 5. Conclusions

In summary, it is long accepted that PVD is a heterogenic pain condition that may represent different disorders. The findings of this study suggest that tactile hypersensitivity in different vestibular anatomic regions, i.e., anterior and posterior, and observation of additional nociceptive markers can lead to clinical subgrouping of women with insertional dyspareunia and vestibular tenderness. The roots of such typology has been partially proposed previously [16,17,19], but empiric evidence for such multifaceted manifestation may facilitate the construction of an algorithm that enables the allocation into subgroups and advance researchers and clinicians. Further investigation should focus on the role of inflammation, neurologic, psychosexual features, and modulatory alteration of nociceptive processing in order to address whether such classification allows for better personalized medicine and improve treatment outcomes.

## Figures and Tables

**Figure 1 jcm-09-02023-f001:**
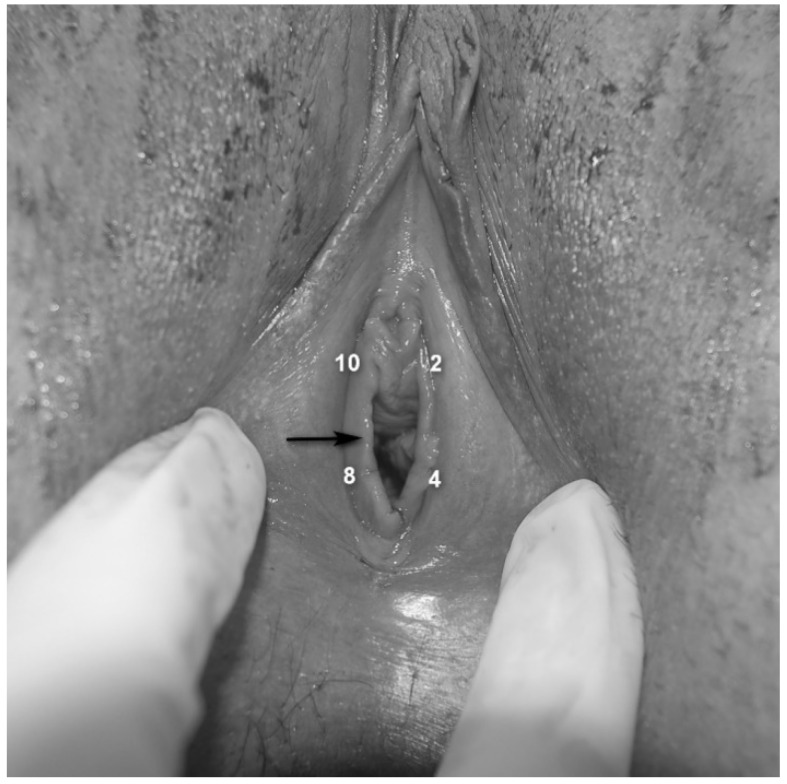
The vulvar vestibule and hymenal ring (black arrow). The assessed vestibular anatomic locations are marked by the numbers 2,4,8 and 10. The hymenal ring is intact albeit previous vaginal penetration and provokes severe pain with a gentle lateral stretching with 2 fingers.

**Figure 2 jcm-09-02023-f002:**
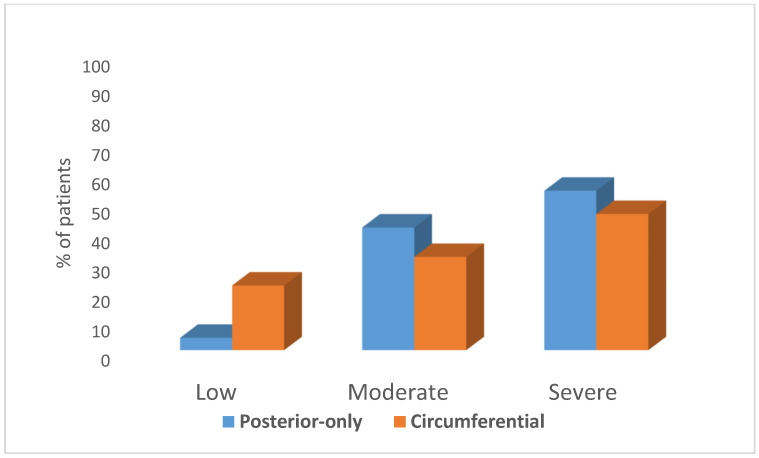
Degrees of pelvic floor hypertonicity according to vestibular tenderness location. Percentage of women with mild, moderate and severe degrees of hypertonicity, according to vestibular tenderness location (circumferential or posterior-only). Those with Posterior-only vestibular tenderness had a higher incidence of moderate and severe hypertonicity in comparison to those with Circumferential vestibular tenderness.

**Figure 3 jcm-09-02023-f003:**
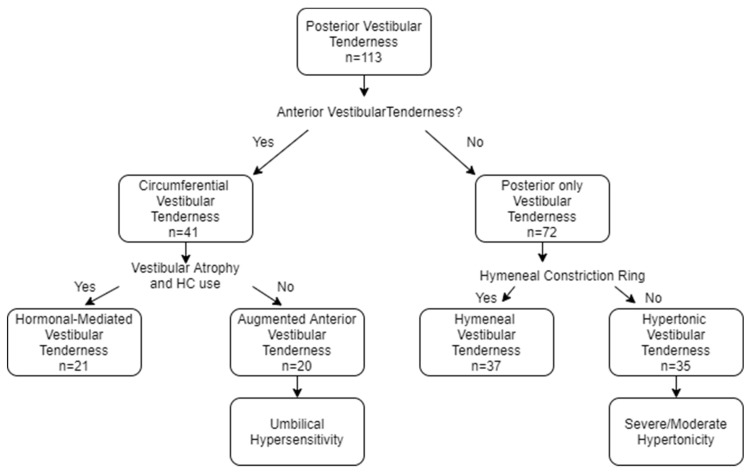
Construction of four subgroups according to vestibular tenderness localization and clinical findings.

**Figure 4 jcm-09-02023-f004:**
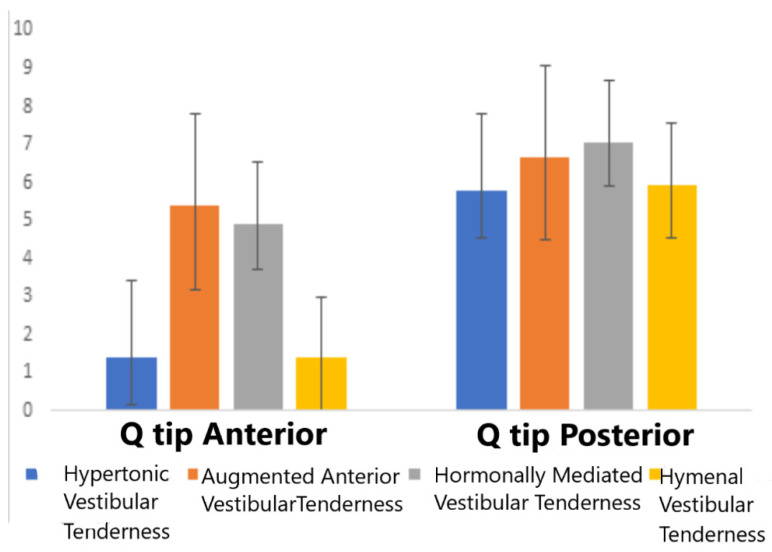
Group comparison of pain intensity scores obtained by Q-tip stimulation at the anterior and posterior vestibule in the four subgroups.

**Table 1 jcm-09-02023-t001:** Sociodemographic characteristics of the study population.

	Mean	Range
Age	26.2 ± 4.1	18–40
Married/in a committed relationship	82 (73.2%)	
Duration of Dyspareunia symptoms (years)	4.1 ± 3.4	4 months–13
Nullipara Para	105 8	
Education (years)	14.23 ± 2.1	11–21
Religiosity	Secular 84 (74.3%) Religious 14 (12.4%) Orthodox 15 (13.3%)	

**Table 2 jcm-09-02023-t002:** Comparison of Circumferential vs. Posterior-only vestibular tenderness subgroups.

	Circumferential Vestibular Sensitivity (*n* = 41)	Posterior-Only Vestibular Sensitivity (*n* = 72)	*p* Value
Vestibular mucosal atrophy	63.4%	20.8%	<0.001
Hormonal Contraceptive use	14.1%	13.6%	NS
Umbilical pain hypersensitivity	46.3%	18.1%	0.001
Rigid hymen	0%	51%	<0.001
Pain intensity during intercourse	8.2 ± 1.5	7.7 ± 1.8	NS
Pain evoked by deep muscle palpation	6.4 ± 2.3	6.3 ± 1.7	NS
Primary PVD	13.5%	13.3%	NS

**Table 3 jcm-09-02023-t003:** Regression model to define predictors for pain variability in the anterior vestibule.

	Unstandardized Coefficients	Coefficients Std. Error	Coefficients Beta	t	*p*
Degree of muscle tonus	0.361	0.436	0.081	0.828	0.410
Pain during intercourse	0.359	0.123	0.276	2.914	0.004
Pain evoked by deep palpation	−0.044	0.114	−0.041	−0.386	0.701
Umbilical sensitivity	1.366	0.4192	0.283	3.262	0.002
Vestibular atrophy	1.140	0.391	0.251	2.917	0.004

**Table 4 jcm-09-02023-t004:** Regression model to define predictors for pain variability in the posterior vestibule.

	Unstandardized Coefficients	Coefficients Std. Error	Coefficients Beta	t	*p*
Degree of muscle tonus	0.203	0.361	0.053	0.368	0.714
Pain during intercourse	0.497	0.102	.440	4.867	0.000
Pain evoked by deep palpation	0.162	0.094	0.173	1.719	0.089
Umbilical sensitivity	0.608	0.347	0.145	1.753	0.083
Vestibular atrophy	0.119	0.324	0.030	0.368	0.714

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
