# Peer review of "Vestibular Anatomic Localization of Pain Sensitivity in Women with Insertional Dyspareunia: A Different Approach to Address the Variability of Painful Intercourse"

_jcm, 2020, doi:10.3390/jcm9072023_

Round 1
Reviewer 1 Report
Dear authors
Thank you for addressing this important topic. The data shown are important and can have a significant impact in terms of future clinical and investigation approach of vestibulodynia.
Vulvodynia has one problem in its definition: once you establish a cause or an explanation, it no longer is vulvodynia... I think that's the most difficult thing about your approach.
In order not to conflict with current definitions, I'd suggest to refer to vestibular pain and putting the focus on possible causes for it. In the end you can conclude that your approach led to an etiological explanation for the pain, avoiding the concept of PVD and that it is likely to lead to a better clinical approach.
Some other comments:
Line 23 – recommend not considering umbilical pain/hypersensitivity as a confounder. The rest of situations in the list are local, all potentially associated with pain. Umbilical pain has been described as common in vulvodynia, but the same is true, for instance, for glossodynia (that can be just a “marker”)
Line 26 - “anterior”, rather than “upper”
Line 33 and 34 – idem
Line 40 – or attempt
Line 48/49 – consider the vestibule as part of the vulva, rather than an independent structure between the vulva and vagina
Line 47/51 – or other causes. For instance, infection
Line 61 – Would not stress too much the Friederich criteria: for instance, most authors do not consider red spots these days as part of the entity
Line 75 – anterior
Line 92 – idem
Line 112 – mean calculation may raise some doubts: one single spot reported as 3 would qualify for the study? Or there were a minimum number?
125 – per definition, this atrophy would exclude the diagnosis of vulvodynia
151/6 – not easy to assure the pain was not attributable to vulvodynia
Line 165 – but with the need of a mean score higher or equal to 3? It needs further explanation
Table 1 – religion: all Jewish?
Line 211/2 – I’m a bit reluctant about this point: if this atrophy is reversed (sometimes by just stopping HC), it does not really fit the definition of vulvodynia. I think a remark should be made about this point (“weak points” section)
Line 250 – “possible hormonal” would keep it in the safe side
Figure 3 – I like this figure, but I see it more as a differential diagnosis approach rather than a way to subgroup vulvodynia
287 – avoid the “as expected”, since this is still the results section
Author Response
We thank the reviewer for the insightful critiques that improved our manuscript.
Vulvodynia has one problem in its definition: once you establish a cause or an explanation, it no longer is vulvodynia... I think that's the most difficult thing about your approach. In order not to conflict with current definitions, I'd suggest to refer to vestibular pain and putting the focus on possible causes for it. In the end you can conclude that your approach led to an etiological explanation for the pain, avoiding the concept of PVD and that it is likely to lead to a better clinical approach.
We completely agree with this comment, and therefore, in order to avoid this contradiction, we used the phrases "vestibular tenderness" and "insertional dyspareunia" throughout the manuscript. We used the term PVD only when we referred to published papers and common definitions. In lines 89-91 we described PVD: "Since "PVD" is defined in the literature as vestibular pain of unknown etiology and the aim of this study was to recognize possibly neglected causes, we used the terms "insertional-dyspareunia" and "vestibular-tenderness" instead."
In the discussion we also included the following sentence: “Furthermore, it should be noted that this study focused on possible etiological explanations for vulvar pain, not related exclusively to PVD. (lines 436-8).”
Some other comments:
Line 23 – recommend not considering umbilical pain/hypersensitivity as a confounder. The rest of situations in the list are local, all potentially associated with pain. Umbilical pain has been described as common in vulvodynia, but the same is true, for instance, for glossodynia (that can be just a “marker”)
We thank the reviewer for this comment; however, we think that this confounder should be considered for the following reasons: 1. The vestibule and the umbilicus have a common endodermal embryological origin, and therefore, may represent a "local" mechanism. 2. This is a concrete, measurable parameter, and is experimentally detectable by local pressure (as opposed to other markers associated with pain which are more difficult to verify and rely on patient's self-report) and 3. We actually found difference in the characteristics of the women using the parameter of umbilical hypersensitivity, as shown in the Results section. This supports our assumption that umbilical hypersensitivity is a confounder in the etiology of vulvar pain.
Line 26 - “anterior”, rather than “upper”
We thank the reviewer for this comment and replaced "upper" with "anterior" throughout the manuscript.
Line 33 and 34 – idem
We thank the reviewer for this comment and replaced "upper" with "anterior" throughout the manuscript.
Line 40 – or attempt
We thank the reviewer for this observation and we added "or penetration attempt".
Line 48/49 – consider the vestibule as part of the vulva, rather than an independent structure between the vulva and vagina
We thank the reviewer for this comment. As the article is geared for readers whom are not necessarily familiar with vulvar anatomy, we wanted to define the vestibule. According to your suggestion, we changed the sentence into "of the vulvar vestibule, a part of the vulva, located adjacent to the vagina".
Line 47/51 – or other causes. For instance, infection
We appreciate this comment and we added the sentence: "Insertional dyspareunia may result from various causes, including infections and inflammation"
Line 61 – Would not stress too much the Friederich criteria: for instance, most authors do not consider red spots these days as part of the entity
We thank the reviewer for this criticism. We did not include erythema or red-spots in the description nor in the inclusion criteria. We deleted the words "by Friedrich" and kept the following sentence: …"provoked-pain on vestibular touch, marked tenderness to cotton swab palpation of the vulvar vestibule and exclusion of other, identifiable causes."
Line 75 – anterior, Line 92 – idem-
We replaced "upper" with "anterior" throughout the manuscript.
Line 112 – mean calculation may raise some doubts: one single spot reported as 3 would qualify for the study? Or there were a minimum number?
We apologize for the confusing phrasing. Inclusion criteria were: self-reported sexual pain intensity ≥3 in the 0-10 Visual Analogue Scale (VAS) or complete avoidance of penetrative intercourse due to pain severity, as well as pain scores >1 in response to Q-tip stimulation.
In order to better clarify this topic, we deleted the word "mean" and replaced it with "self-reported" (this is now line 117).
We added the description regarding Q-tip test is in lines 119-120: "tenderness localized within the vestibule on examination, with pain scores >1 in response to Q-tip stimulation"
125 – per definition, this atrophy would exclude the diagnosis of vulvodynia
We completely agree. We, therefore, did not define the patients as having PVD but rather having "vestibular tenderness" and "insertional dyspareunia". We also explained in detail the clinical presentation of vestibular atrophy, which differs from "vaginal atrophy" in lines 130-133.
151/6 – not easy to assure the pain was not attributable to vulvodynia
The reviewer's comment raises very important issue, however, we do not fully agree with this statement. The described evaluation process differed between pain originating from mucosal vestibular tenderness, hymenal-stretching tenderness and muscular tenderness. As is described in the Results section, hymenal component was not universal and allowed characterization of those with both posterior-only vestibular tenderness and hymenal-tenderness as opposed to other patients, who did not present this feature. As a result, these patients cannot be considered as having vulvodynia.
Line 165 – but with the need of a mean score higher or equal to 3? It needs further explanation
Please see our explanation above (line 112).
Table 1 – religion: all Jewish?
We thank the reviewer for this question. We asked the patients two questions, 1. What is their religion (Jewish, Muslim, Christian or Other)- all but two respondents were Jewish, one did not answer and one chose "Other", and 2. Level of Religiosity: orthodox, religious or secular.
Line 211/2 – I’m a bit reluctant about this point: if this atrophy is reversed (sometimes by just stopping HC), it does not really fit the definition of vulvodynia. I think a remark should be made about this point (“weak points” section)
We completely agree. Therefore, we did not define the patients as having PVD but rather having "vestibular tenderness" and "insertional dyspareunia". A remark was also added, lines 436-8.
Line 250 – “possible hormonal” would keep it in the safe side
We agree that at this point, this is not an established connection but a possibility, however, adding "possible" whenever it is written "Hormonally-mediated vestibular-tenderness" in the manuscript is cumbersome, so the following sentence was added: " …possibly experiencing hormonally-mediated vestibular sensitivity, were classified as Hormonal-mediated …"
Figure 3 – I like this figure, but I see it more as a differential diagnosis approach rather than a way to subgroup vulvodynia
We thank the reviewer for this comment and changed the figure (deleted "PVD" and replaced it with "vestibular tenderness"), so it represents the diagnostic approach but does not mention the word "vulvodynia".
287 – avoid the “as expected”, since this is still the results section
According to this comment, we deleted the words "as expected".
Reviewer 2 Report
This appears to be a sound study and well-needed in this little studied area of research.
I have one major issue with the design of this study, which I would like the authors to appropriately address:
1) Under Materials and Methods, the authors state that participants were excluded if they had diagnoses of psychiatric disorders. While neuromodulatory medication exposure is an understandable confound (particularly in cases of a small N, which might preclude addressing it in statistical analysis), excluding women with psychiatric diagnoses smacks of a concerning trend in medical research that continues to treat female-related disorders from an unfortunate Freudian perspective and undermines the patient experience. It also ignores the fact that the literature suggests women with psychiatric diagnoses may have higher rates of physical disorders such as dyspareunia, suggesting that the authors may have overlooked an important subtype of this condition, which they are purporting to study.
I am aware that this is an irreparable issue at this stage of the study and I would not prevent the dissemination of these findings because of it as I consider them very useful in this area of biomedical research. However, I would like to see this very significant oversight acknowledged as well as a pledge to avoid such errors in future studies of this nature. This is especially important considering the history of this particular disorder and its previous inclusion within the DSM as a "sexual dysfunction."
I work with several clinical populations, predominantly women, who have medical syndromes that coincide not only with psychiatric but also neurodevelopmental disorders, the latter strongly indicating that the behavioral manifestations are yet one more symptom of a larger disease, not the direct cause of the physical phenomena themselves. We really need to move past this antiquated notion and also remember that correlation does not equal causation. Simply because a physical symptom accompanies a psychiatric/psychological/behavioral one does not mean the former is a result of the latter.
One other minor issue:
1) Were there any data collected in regards to endocrine disorders in these participants? Or were they removed as per exclusion criteria (major health concerns)? Common endocrine disorders, such as polycystic ovary syndrome, are often treated with HC and may be a subgroup worthwhile addressing. If they were excluded from the analysis, it would be worthwhile mentioning that alongside "(cardiovascular, diabetes etc.)".
Author Response
This appears to be a sound study and well-needed in this little studied area of research.
I have one major issue with the design of this study, which I would like the authors to appropriately address:
1) Under Materials and Methods, the authors state that participants were excluded if they had diagnoses of psychiatric disorders. While neuromodulatory medication exposure is an understandable confound (particularly in cases of a small N, which might preclude addressing it in statistical analysis), excluding women with psychiatric diagnoses smacks of a concerning trend in medical research that continues to treat female-related disorders from an unfortunate Freudian perspective and undermines the patient experience. It also ignores the fact that the literature suggests women with psychiatric diagnoses may have higher rates of physical disorders such as dyspareunia, suggesting that the authors may have overlooked an important subtype of this condition, which they are purporting to study.
I am aware that this is an irreparable issue at this stage of the study and I would not prevent the dissemination of these findings because of it as I consider them very useful in this area of biomedical research. However, I would like to see this very significant oversight acknowledged as well as a pledge to avoid such errors in future studies of this nature. This is especially important considering the history of this particular disorder and its previous inclusion within the DSM as a "sexual dysfunction."
I work with several clinical populations, predominantly women, who have medical syndromes that coincide not only with psychiatric but also neurodevelopmental disorders, the latter strongly indicating that the behavioral manifestations are yet one more symptom of a larger disease, not the direct cause of the physical phenomena themselves. We really need to move past this antiquated notion and also remember that correlation does not equal causation. Simply because a physical symptom accompanies a psychiatric/psychological/behavioral one does not mean the former is a result of the latter.
We thank the reviewer for this significant comment and would like to clarify. The exclusion criterion of "psychiatric disorders" was intended to exclude women presenting with unstable psychiatric disorders. We added to line 123: "diagnosis of an unstable psychiatric disorder". This exclusion criteria was a result of the demanding nature of the study (which only its initial appointment was described in this manuscript) as well as our objective to recruit representative population with dyspareunia. In addition, our approach included collection of multiple psychologic and cognitive components using the following questionnaires: stress was obtained using the Perceived Stress Scale (PSS), neuroticism via the NEO PI-R as well as somatization, anxiety and depression by the Brief Symptom Inventory-18 (BSI-18). It should be noted that these questionnaires are not diagnostic. However, the self-reported responses revealed 12% of participants had a history of diagnosed depression and/or anxiety, while 21% reported current perceived depressive symptoms, 26% reported that they experienced anxiety and 10% reported obsessive-compulsive symptoms, but these conditions were not defined as anxiety, OCD or other psychiatric disorders.
While we excluded women with mixed vulvodynia, breastfeeding, and those with concurrent vulvar dermatosis (see results), no one was excluded due to a psychiatric disorder.
One other minor issue:
Were there any data collected in regards to endocrine disorders in these participants? Or were they removed as per exclusion criteria (major health concerns)? Common endocrine disorders, such as polycystic ovary syndrome, are often treated with HC and may be a subgroup worthwhile addressing. If they were excluded from the analysis, it would be worthwhile mentioning that alongside "(cardiovascular, diabetes etc.)".
We thank the reviewer for this comment. Actually, additional data was collected, including endocrine disorders, comorbid pain syndromes, allergies, recurrent infections and many more. We plan to include these data in subsequent papers.
Reviewer 3 Report
The authors submit an original badly needed manuscript on insertional dyspareunia (PVD). This extremely common entity responsible for much pain and suffering worldwide is poorly studied. Clinicians (usually gynecologist) are largely without skills to evaluate PVD, develop a differential diagnosis and rational treatment plan. While multiple causes of PVD are recognized, several investigators have recognized that more than one pathophysiologic mechanism exists resulting in clinical subgroups of women with PVD. However, distinguishing the subgroups is difficult and largely unstudied, hence even committed, well-intentioned clinicians struggle to separate subgroups and determine causation and hence therapy descends into empiricism.
With this background, the authors attempt to add consistent clinical experience (single practitioner) using relatively simplistic confounding variables based upon physical diagnosis to separate women with posterior-only vestibular tenderness from those with circumferential-vestibular-tenderness, hence allowing enhanced therapeutic intervention…. all starting from “location of vestibular tenderness”, together with evaluating additional pain-related features.
1). Exclusion factor-spontaneous or mixed vulvodynia please explain both.
2). While the statistical analyses allow for partial subgroup identification, it would be advantageous to document how this approach facilitates decision making with regard to treatment selection and treatment outcome. This clearly is the next step translating and transforming this technique of anatomic localization of pain sensitivity into widespread therapeutic advantage.
Author Response
The authors submit an original badly needed manuscript on insertional dyspareunia (PVD). This extremely common entity responsible for much pain and suffering worldwide is poorly studied. Clinicians (usually gynecologist) are largely without skills to evaluate PVD, develop a differential diagnosis and rational treatment plan. While multiple causes of PVD are recognized, several investigators have recognized that more than one pathophysiologic mechanism exists resulting in clinical subgroups of women with PVD. However, distinguishing the subgroups is difficult and largely unstudied, hence even committed, well-intentioned clinicians struggle to separate subgroups and determine causation and hence therapy descends into empiricism.
With this background, the authors attempt to add consistent clinical experience (single practitioner) using relatively simplistic confounding variables based upon physical diagnosis to separate women with posterior-only vestibular tenderness from those with circumferential-vestibular-tenderness, hence allowing enhanced therapeutic intervention…. all starting from “location of vestibular tenderness”, together with evaluating additional pain-related features.
We thank the reviewer for acknowledging the potential importance of our study.
1). Exclusion factor-spontaneous or mixed vulvodynia please explain both.
We thank the reviewer and we have explained it in the manuscript (lines 125-8):
6) diagnosis of spontaneous (i.e., pain arising without any provoking physical contact) or mixed vulvodynia (i.e., a combination of spontaneous vulvodynia and pain provoked by physical contact)
2). While the statistical analyses allow for partial subgroup identification, it would be advantageous to document how this approach facilitates decision making with regard to treatment selection and treatment outcome. This clearly is the next step translating and transforming this technique of anatomic localization of pain sensitivity into widespread therapeutic advantage.
We completely agree. As the prospective nature of the study allowed as long-term follow up, we collected additional data, including comorbid pain syndromes, medical background, cognitive and psychologic data and follow-up in regard to personalized treatment, according to the suggested subgrouping. We plan to include these data in subsequent papers.